# Identification of the WUSCHEL-Related Homeobox (*WOX*) Gene Family, and Interaction and Functional Analysis of TaWOX9 and TaWUS in Wheat

**DOI:** 10.3390/ijms21051581

**Published:** 2020-02-26

**Authors:** Zheng Li, Dan Liu, Yu Xia, Ziliang Li, Doudou Jing, Jingjing Du, Na Niu, Shoucai Ma, Junwei Wang, Yulong Song, Zhiquan Yang, Gaisheng Zhang

**Affiliations:** College of Agronomy, Northwest A&F University, National Yangling Agricultural Biotechnology & Breeding Center, Yangling Branch of State Wheat Improvement Centre, Wheat Breeding Engineering Research Center, Ministry of Education, Key Laboratory of Crop Heterosis of Shaanxi Province, Yangling 712100, China; lizheng9045@nwafu.edu.cn (Z.L.); zoeyld@nwafu.edu.cn (D.L.); xiayu325@nwafu.edu.cn (Y.X.); lzl5125@nwafu.edu.cn (Z.L.); jdd1005@nwafu.edu.cn (D.J.); dou0321@nwafu.edu.cn (J.D.); niuna@nwsuaf.edu.cn (N.N.); mashoucai@sohu.com (S.M.); wjw@nwsuaf.edu.cn (J.W.); sylbl1986@163.com (Y.S.); zhqyang6628@nwsuaf.edu.cn (Z.Y.)

**Keywords:** *WOX* gene family, flower development, interaction analysis, *TaWOX9*, wheat

## Abstract

The WUSCHEL-related homeobox (WOX) is a family of plant-specific transcription factors, with important functions, such as regulating the dynamic balance of division and differentiation of plant stem cells and plant organ development. We identified 14 distinct *TaWOX* genes in the wheat (*Triticum aestivum* L.) genome, based on a genome-wide scan approach. All of the genes under evaluation had positional homoeologs on subgenomes A, B and D except TaWUS and TaWOX14. Both *TaWOX14a* and *TaWOX14d* had a paralogous copy on the same genome due to tandem duplication events. A phylogenetic analysis revealed that *TaWOX* genes could be divided into three groups. We performed functional characterization of *TaWOX* genes based on the evolutionary relationships among the *WOX* gene families of wheat, rice (*Oryza sativa* L.), and *Arabidopsis.* An overexpression analysis of *TaWUS* in *Arabidopsis* revealed that it affected the development of outer floral whorl organs. The overexpression analysis of *TaWOX9* in *Arabidopsis* revealed that it promoted the root development. In addition, we identified some interaction between the TaWUS and TaWOX9 proteins by screening wheat cDNA expression libraries, which informed directions for further research to determine the functions of TaWUS and TaWOX9. This study represents the first comprehensive data on members of the WOX gene family in wheat.

## 1. Introduction

Homeobox (HB) proteins were first discovered in *Drosophila*. They belong to a large family of transcriptional factor proteins, characterized by the presence of a short stretch of amino acids (60–66 residues). These amino acids fold into a DNA-binding domain termed the homeodomain, which is encoded by the HB DNA sequence [1]. The HB proteins have been identified in all the eukaryotic organisms evaluated [2]. In higher plants, many homeodomain (HD)-containing transcriptional factor proteins have been identified in both monocots and dicots [3], of which KNOTTED1 is the first HD-containing protein to be identified [4]. The HB protein superfamily is classified into six families. These classifications include homeodomain-leucine (HD-Zip); plant homeodomain (PHD)-finger; BELL; zinc finger-homeodomain (ZF-HD); WUSCHEL (WUS)-related homeobox (WOX); and KNOTTED1-like-homeobox (KNOX) [2,3].

In plants, members of the *WOX* gene family play important roles in multiple development processes, including embryonic development, embryonic polarization, maintenance of meristematic stem cells, development of lateral organs, seed formation, and regeneration of isolated tissues and organs [5]. The *WOX* gene family has been identified in many plants. The model eudicot plant *Arabidopsis* contains 15 WOX proteins, which are classified into three clades based on evolutionary relationships: a modern/WUS clade, an intermediate clade, and an ancient clade [1,6]. The modern/WUS clade contains AtWUS and AtWOX1-7. The intermediate clade includes AtWOX8, 9, 11, and 12. The ancient clade contains AtWOX10, 13, and 14. Several members of *Arabidopsis* WOX genes have been shown to be essential for embryonic patterning, stem cell maintenance in the shoot apical meristem (SAM) and root apical meristem (RAM), and organ formation [6]. The AtWUS protein is involved in maintaining stem cell homeostasis in the SAM at all developmental stages, and in the wus mutant, SAM maintenance is disrupted both embryonically and postembryonically [7,8]. The AtWOX2 protein is expressed in the egg cell and zygote, and is required for determining cell fate in the apical and basal cell lineages of developing embryos [9,10]. The PRESSED FLOWER1 (PRS1/AtWOX3) protein is involved in the initiation and development of lateral organs, as it recruits founder cells from all meristem layers in *Arabidopsis* [11]. As a homolog of AtWUS, AtWOX5 is specifically expressed in the quiescent center of the root and is essential for stem cell maintenance in the RAM via a negative feedback signal provided by CLE40 [12]. The PRETTY FEW SEEDS2 (PFS2/AtWOX6) protein is abundantly expressed in developing ovules and plays a key role during ovule patterning by regulating both cell proliferation of the maternal integuments and differentiation of the megaspore mother cell [13]. The STIMPY (STIP/AtWOX9) protein is required for the growth of the vegetative SAM, can maintain cell division, and prevent premature differentiation in tissues of the SAM, some other aerial organs, and the root [14].

There are at least 13 WOX proteins in rice (*Oryza sativa* L.) The MONOCULM 3/TILLERS ABSENT 1/STERILE AND REDUCED TILLERING 1 (MOC3/TAB1/SRT1) protein, an ortholog of WUSCHEL, is required for tiller bud formation and female fertility in rice, and has a complex relationship with cytokinins [15,16]. The MOC1 and MOC3 proteins physically interact to regulate tiller bud outgrowth by upregulating the expression of FLORAL ORGAN NUMBER1 (FON1), the orthologs of CLAVATA1 in rice, in a similar manner to CLV signaling in *Arabidopsis* [17]. The rice narrow leaf2 and narrow leaf3 loci encode an identical OsWOX3A (OsNS) transcriptional activator, orthologs to NARROW SHEATH1 (NS1) and NS2 in maize (*Zea mays* L.) and PRESSED FLOWER in *Arabidopsis*. The OsWOX3A protein is involved in leaf, spikelet, tiller, and lateral root development [18]. Barley *NARROW LEAFED DWARF1 (NLD1)* encodes a WOX3, an ortholog of the maize *NS* genes. Studies have shown that NLD1 plays pivotal role in the increase of organ width and in the development of marginal tissues in lateral organs in barley [19]. Both OsWOX6 and OsWOX11 are expressed asymmetrically in response to auxin, to connect gravitropic responses and control the tiller angle in rice [20]. The OsWOX11 protein is also able to activate the emergence and growth of the crown root, and the overexpression of OsWOX11 leads to the formation of multiple pistils in rice [21]. Despite these findings, only a few systematic studies have been conducted on the *WOX* gene family in bread wheat (*Triticum aestivum* L.)

Bread wheat has an allo-hexaploid genome consisting of three closely related subgenomes (AABBDD). It is proposed to have originated from two polyploidization events; first, a tetraploidization from the hybridization between wild Triticum urartu Tumanian ex Gandilyan (AA) and an undiscovered species of the Aegilops speltoides Tausch lineage (BB), and second, a hexaploidization from the hybridization between a descendant of this original tetraploid hybrid (AABB) and the wild diploid Aegilops tauschii Coss (DD) [22]. These two polyploidization events are some 0.5 million years before present (ybp) and some 10,000 ybp, respectively. Because of the complexity of the wheat genome, it has hindered the study of wheat gene function. However, rice is the most widely studied model plant in monocotyledons, providing a reference for the study of other monocotyledons such as wheat.

In this study, we analyzed the *WOX* gene family in wheat. We identified 14 genes of the WOX gene family in wheat, based on a genome-wide scan approach, and predicted their functions by combining the analysis of the phylogenetic tree with that of the expression patterns. We also studied the potential roles of *TaWUS* and *TaWOX9* in the development of the flower and root, by identifying their interacting proteins and conducting an overexpression analysis of both genes in *Arabidopsis*. This study provides further insight into the structure and function of the *WOX* gene family in wheat, and is useful for their further functional studies.

## 2. Results

### 2.1. Identification and Phylogenetic Analysis of WOX Genes in Wheat

To identify all the *WOX* genes in wheat, we used known WOX protein sequences of *Arabidopsis* and rice as queries to search the database of the wheat reference genome through homology alignment and iterative search. This resulted in the identification of 44 wheat *WOX* genes (Appendix A). Wheat is an allohexaploid plant with three subgenomes, A, B, and D. Most genes in the complex genome of wheat contain homoeologs and paralogs. Considering these factors, the 42 genes identified can be integrated into 14 distinct wheat genes (Appendix A), which is similar to the number of WOX genes found in other plants [23]. For research convenience, these TaWOX genes were named *TaWUS-TaWOX14* based on their sequence homology and evolutionary relationships with members of the rice *WOX* family (Figure 1A). The name of the *TaWUS* gene is based on the orthology between TaWUS and AtWUS. The physical positions of 42 *TaWOXs* were found on 15 chromosomes of wheat, being absent on chromosomes 6A, 6B, 6D, 7A, 7B, and 7D (Appendix A).

The number of WOX genes on different chromosomes ranged from one to seven. One gene was located on chromosomes 4A, 4B, and 4D, two genes on chromosomes 1A, 1B, 1D, 2A, 2B, 2D, 5A, 5B, and 5D, six genes on chromosome 3B, and seven genes on chromosomes 3A and 3D. Owing to tandem duplication events, *TaWOX14a and TaWOX14d* had a paralogous copy on chromosomes 3A and 3D, respectively (Appendix A). The putative WOX proteins in wheat (TaWOXs) ranged from 208 to 523 amino acids in length (Appendix A). The predicted molecular weights of TaWOX proteins ranged from 24.2 kDa (TaWOX9b) to 53.7 kDa (TaWOX7a) (Appendix A). The predicted pI ranged from 5.79 (TaWOX8a and TaWOX8b) to 10.41 (TaWOX5b) (Appendix A).

A phylogenetic tree of WOX proteins from wheat, rice, and *Arabidopsis* was generated, based on the full WOX amino acid sequences (Figure 1A). Similar to the HD sequence-based phylogeny of known plant WOX proteins, the 14 TaWOXs were also divided into three major branches: the modern/WUS clade, the intermediate clade, and the ancient clade, which were similar to the classification of the *WOX* gene family in *Arabidopsis* and rice (Figure 1A). The modern/WUS clade contained 24 *TaWOX* genes: *TaWUS a*; *TaWOX2*
*(a*, *b*, *d)*; *TaWOX3*
*(a*, *b*, *d)*; *TaWOX4*
*(a*, *b*, *d)*; TaWOX5 (a, b, d); TaWOX9 (a, b, d); TaWOX13 (a, b, d); and TaWOX14 (a1, a2, b, d1, d2). The intermediate clade consisted of 15 *TaWOX* genes: *TaWOX6*
*(a*, *b*, *d)*; *TaWOX7*
*(a*, *b*, *d)*; *TaWOX10*
*(a*, *b*, *d)*; *TaWOX11*
*(a*, *b*, *d)*; and *TaWOX12*
*(a*, *b*, *d)*. The ancient clade contained only *TaWOX8*
*(a*, *b*, *d)*.

Our phylogenetic analysis unambiguously established the orthologous relationship between TaWOX and OsWOX proteins, as well as some differences between TaWOX and AtWOX proteins (Figure 1A). All the TaWOX proteins were related to 13 OsWOX proteins, but no TaWOX proteins were closely related to AtWOX1, AtWOX6, AtWOX8, AtWOX9, AtWOX10, or AtWOX14 in the phylogenetic tree.

### 2.2. Multiple Sequence Alignment and Sequence Features of TaWOX Proteins

The *Arabidopsis* WUS protein contains three typical functional domains: the homeodomain (HD), which is the most prominent and defining feature of the family, the WUS box, and the EAR-like motif. To reveal the three functional domains in the wheat WOX gene family, we performed sequence analysis on wheat WOX proteins via multiple sequence alignment and the prediction from the Simple Modular Architecture Research Tool (SMART) software. The results showed that the similarity between the protein sequences of homoeologs and paralogs in the TaWOX gene family is very high, especially for these conserved domains (Appendix A). Therefore, only one of the homoeologous copies was selected in the analysis of function domain below. The results also showed that each group classified by phylogenetic analysis shared similar domain compositions; however, some differences were also noted between different subgroups. As shown in Figure 1B, HDs were present in all TaWOXs. The WUS-box motif existed in the modern/WUS group alone. The EAR motif was present only in TaWOX9 and TaWUS, which belong to the modern/WUS group.

Multiple sequence alignment of the HD sequences of 42 WOX proteins was conducted (Figure 2A). The results showed that the distribution of conserved amino acids among the HDs of TaWOX proteins was extremely similar to those of AtWOX proteins and OsWOX proteins. Previous studies have reported the presence of 11 conserved amino acids in the HD region, including Q, L, and Y in helix 1, and I, V, W, F, N, K, R, and R in helix 3 [23]. These amino acids are also conserved in the HD of TaWOX proteins. In addition to these conserved amino acids that were identified, we observed additional conserved residues, including P, I, and L in helix 2; F and Q in helix 3; and G in the turn among these TaWOX members (Figure 2A). The presence of an extra amino acid in the loop sequences of TaWUS, AtWUS, and OsWOX1 (also called OsWUS), compared to the loop sequences of other WOX proteins, is noteworthy. Moreover, the last amino acid of Helix 1 in TaWUS, AtWUS, and OsWOX1 is Y, which differs from those of other WOX proteins. Multiple sequence alignments of the WUS-box motif indicated that the conserved WUS-box motifs were mainly “TLXLFP” (where X represents any amino acid) (Figure 2B); however, there were exceptions. For instance, the first amino acids of the WUS-box motif of TaWOX13, TaWOX14, and TaWUS were L, L and E, respectively. Multiple sequence alignment revealed that the EAR motifs of TaWUS and TaWOX9 were highly similar to or identical with the conserved EAR motifs in the AtWUS, AtWOX5, AtWOX7, OsWOX1, and AtWOX9 proteins (Figure 2C).

### 2.3. Expression Profile of TaWOX Genes in Wheat

To dissect the expression patterns of TaWOX genes in various tissues, nine diverse tissues, i.e., the young spike, sheath, stem, seed, root, ovary, leaf, glume, and anther, were included in the analysis. The results showed that TaWOX genes had different expression profiles in different tissues and organs at different stages. In comparison, TaWOX4, TaWOX9, TaWOX10, TaWOX11, and TaWOX13 had higher expression levels tissues in the vegetative stage (Figure 3). TaWOX2, TaWOX5, TaWOX8, TaWOX12, and TaWOX14 had higher expression levels in tissues in the reproductive stage (Figure 3; Appendix A). TaWUS, TaWOX3, TaWOX6, and TaWOX7 had considerable expression levels in the tissues at both stages (Figure 3). Among them, TaWUS, TaWOX2, TaWOX8, and TaWOX14 showed the highest expression levels in the development of young spikes (Figure 3; Appendix A). TaWOX9, TaWOX10, and TaWOX11 were preferentially expressed in the roots, suggesting that they may play an important role in root development (Figure 3; Appendix A). TaWOX5, TaWOX12, and TaWOX13 were preferentially expressed in developing grains, anthers and leaves, respectively (Figure 3), suggesting that their functions may be closely related to the development of those tissues. TaWOX3 showed higher expression in leaves, roots and young spikes (Figure 3). TaWOX4 had relatively high expression in leaves, sheaths, stems and roots in the vegetative stage (Figure 3). TaWOX6 was preferentially expressed in sheaths and stems in the vegetative stage and in young spikes and anthers in the reproductive stage (Figure 3). TaWOX7 was expressed in various tissues, especially in anthers, grains, roots, young spikes, and the ovary (Figure 3). It may be involved in basic metabolism during wheat development. The developing young spikes of wheat are filled with a large number of stem cells. As young spikes develop, these stem cells can gradually differentiate into wheat flower organs. Most TaWOX genes, including TaWUS, TaWOX2, TaWOX3, TaWOX6, TaWOX7, TaWOX8, TaWOX11, and TaWOX14, are highly expressed in wheat spikes (Figure 3; Appendix A). This indicates that these genes may be involved in the maintenance of stem cells.

### 2.4. Subcellular Localization and Transactivation Activity of TaWUS and TaWOX9

TaWUS and TaWOX9 were the only two genes among wheat WOX members that contain both the WUS box and EAR-like motif. These two functional domains may confer dual functions to TaWUS and TaWOX9 similar to those of AtWUS. Therefore, we next focused on the functional analysis of these two genes. To ascertain whether they were able to activate transcription, TaWUS and TaWOX9 were each fused with the GAL4 DNA-binding domain (GAL4BD) and tested in yeast, using a reporter construct. The result showed that both TaWUS and TaWOX9 were able to activate the expression of two reporter genes, *HIS3* and *ADE2* in yeast strain Y2H, (Figure 4A), indicating that they were transcriptional activators. Moreover, we also confirmed that the transcription activation regions of TaWUS and TaWOX9 were located at the C-terminus using similar methods. Furthermore, the N-terminus containing the HD region showed no activation capability (Figure 4A). The fluorescent protein-tagging method was used to investigate whether TaWUS and TaWOX9 were nuclear-localized proteins like transcription factors (TFs). The results showed that while green fluorescent protein (GFP) alone presented a dispersed cell distribution (Figure 4B), GFP-tagged TaWUS and TaWOX9 were located in the nucleus, as per their functions as TFs (Figure 4C,D).

### 2.5. Identification of Proteins That Interact with TaWUS and TaWOX9

Subcellular localization and autoactivation assays showed that TaWUS and TaWOX9 are typical transcription factors. Transcription factors often interact with other transcription factors or proteins to form a transcription complex, and then directly or indirectly regulate the expression of downstream genes. Therefore, identifying the interaction proteins of a particular transcription factor and identifying its downstream target genes are required to reveal its function. To understand the functions of TaWUS and TaWOX9 proteins, we tried to identify their interaction partners via yeast two-hybrid screening. Since neither TaWUS-N-BD nor TaWOX9-N-BD had activation capability, we chose them as baits, to conduct two-hybrid screening against the wheat mix cDNA expression libraries. For TaWUS, out of the 60 positive clones, 24 harbored cDNA inserts in the correct reading frame, and were identified as six independent proteins (Figure 5A). The six interaction proteins were identified in at least two independent screens, suggesting that they represent a good TaWUS interacting protein candidate. For TaWOX9, out of the 60 positive clones, only 14 clones contained cDNA inserts in the correct reading frame. These 14 clones were identified as two independent proteins (Figure 5A). Interestingly, one of the two proteins, TaPUB4, interacted with TaWUS too. Among the six interacting proteins of TaWUS, The TaFLX1 was orthologous to AtFLL1 (Appendix A). TaFVE encoded a WD-40 repeat-containing protein, which was orthologous to *Arabidopsis* FVE (Appendix A). TaPUB4 encodes a plant U-box E3 ligase, which was orthologous to AtPUB4 (Appendix A). TaSEP2 and TaSEP3 were orthologous to *Arabidopsis* SEPALLATA2 (SEP2) and SEP3 (Appendix A), respectively. TaBTB18 encodes a tramtrack and broad (BTB) domain scaffold proteins, which was orthologous to AtBT2 (Appendix A). In addition, we also confirmed that OsPUB4 and AtPUB4 could interact with OsWUS and AtWUS in yeast cells, respectively (Appendix A). TaPUB4 could not interact with TaWOX3 and TaWOX8 in yeast cells. (Appendix A).

We further conducted bimolecular fluorescence complementation (BiFC) assays in *Nicotiana benthamiana* leaf cells to detect the interaction between TaWUS and its interacting proteins. The results showed that the interactions between TaWUS and the six proteins took place in the nucleus (Figure 5B–H). Similarly, we used BiFC to verify the interactions between TaWOX9 and its two interaction proteins (Figure 5I,J). The results showed that the interactions between TaWOX9 and both interaction proteins took place in the nucleus.

We also performed tissue expression analysis on the above seven interacting genes. The results showed a high variability in the expression level of the seven genes in various tissues and organs, indicating the diversified functions of the seven genes in wheat growth and development. The seven genes could be detected in young spike, sheaths, stems, grains, roots, ovary, leaves, glumes, and anthers using RT-qPCR (Appendix A). TaPUB4, TaFLXL1 and TaLUXL had higher expression levels in leaves, sheaths, stems and roots in the vegetative stage. TaSEP2 was relatively highly expressed in young spikes. TaFVE showed the highest expression in the ovary, but also higher expression in leaves, sheaths and stems. TaSEP3 had the first three expression levels in anthers, ovary and young spikes. TaBTB18 was relatively highly expressed in leaves, sheaths, and stems, but especially in leaves.

### 2.6. Overexpression Analysis of TaWUS and TaWOX9 in Arabidopsis

To understand the role of TaWUS and TaWOX9 in plant development, their overexpression lines (TaWUS-OE and TaWOX9-OE) were constructed in *Arabidopsis*. Two independent homozygous T3 transgenic lines were used for analysis. Expression levels of TaWUS and TaWOX9 were confirmed using qRT-PCR. TaWUS and TaWOX9 were highly expressed in the two positive lines, but not expressed in the WT (Appendix A). Compared with those of the wild type (WT), the petal number of TaWUS-OE was markedly increased (Figure 6A–D). The number of petals was as high as seven (Figure 6B). The number of sepals of some TaWUS-OE flowers was also increased, compared with the WT (Figure 6D). However, the number of stamens and pistils in the inner whorl remained unchanged in TaWUS-OE (Figure 6A–D). The number of carpels in silique of TaWUS-OE also remained unchanged compared to WT (Appendix A). In addition, TaWUS-OE had a greater number of flower buds than the WT during the same developmental period (Figure 6E,F). Flowering time in TaWUS-OE was also slightly earlier than in the WT (Appendix A). There were more leaves at the branch nodes of TaWUS-OE than WT (Appendix A), indicating that lateral meristem activity is enhanced in TaWUS-OE. As TaWOX9 is expressed mainly in roots, we analyzed the roots of the TaWOX9-OE lines. The roots of TaWOX9-OE were markedly longer than those of the WT, and the lateral roots of TaWOX9-OE were more abundant than those of the WT during the same developmental period (Figure 7).

## 3. Discussion

### 3.1. Overview of the TaWOX Gene Family

The *WOX* gene family is one of the highly conserved gene families in plants. Their function is essential for normal plant development and their mutations can often cause serious defects. Owing to the importance of their function, current research in plants has been both in-depth and extensive. Nevertheless, so far, research on the functions of the *TaWOX* gene in wheat has been very limited, and the *TaWOX* gene family members have not been systematically identified and defined. In this study, we identified 14 distinct *TaWOX* genes from the wheat reference genome. These *TaWOX* genes were distributed on all wheat chromosomes except chromosomes 7 and 6. The number of *TaWOX* members is generally equivalent to that of *WOX* members in other plants. Wheat is an allohexaploid species with a very complex genome. *TaWOX* genes have homoeologs in the A, B, and D subgenomes due to polyploidization. *TaWOX14a* and *TaWOX14d* also have a paralogous copy on the same genome, due to a tandem duplication event. A cluster analysis of AtWOX proteins and OsWOX proteins revealed that TaWOX proteins could be divided into three clades: the modern/WUS clade, the intermediate clade, and the ancient clade. This classification is basically consistent with reported classifications in other plants [23]. TaWOX proteins include the HD, WUS-box motif, and EAR motif. The WUS-box motif exists only in the modern/WUS clades, similar to those in *Arabidopsis* and rice [23]. Among the *Arabidopsis* WOX transcription factor family, WUS and WOX5 contain, apart from the WUS-box motif, a conserved EAR-like motif at the C-terminus, and its main function is to suppress the expression of downstream genes [23]. After further analysis, we found that the C-terminus of TaWUS and TaWOX9 also contained a conserved EAR-like motif. A previous study has shown that *Arabidopsis* WUS is a bifunctional transcription factor that acts as a repressor in stem cell regulation and an activator in floral patterning, and the essential role of the WUS box in all activities of WUS [24]. Therefore, we speculated that because TaWUS and TaWOX9 contain the WUS box and EAR-like motif, they might also play dual roles of activation and inhibition, similar to AtWUS.

### 3.2. Putative Functions of TaWOX Genes Based on Sequence Homology, Gene Expression Pattern and Heterologous Gene Expression

The TaWOX gene family is also highly conserved in terms of the number of genes and the sequence of the functional domain compared to the WOX genes in other plants. Therefore, it is very useful for identifying the function of TaWOX genes via analysis of the evolutionary cluster and tissue-specific expression. For example, TaWOX3 is the putative ortholog of OsWOX3. In rice, WOX3A is expressed in various organs, and OsWOX3A mutations result in narrow leaves and fewer lateral roots [18]. Our tissue expression analysis showed that TaWOX3 was highly expressed in leaves and roots, suggesting that TaWOX3 may perform similar functions to OsWOX3A in the development of leaves and roots. TaWOX11 is the putative ortholog of OsWOX11. In rice, OsWOX11 is involved in the regulation of crown root development [21]. A tissue expression analysis showed that TaWOX11 was preferentially expressed in wheat roots, indicating that it may be involved mainly in wheat root development. In the phylogenetic tree, TaWOX9, AtWOX7, AtWOX5, and OsWOX9 were clustered on a single branch. AtWOX5 and AtWOX7 are reportedly involved in the development of Arabidopsis roots [12,25]. A tissue expression analysis also showed that TaWOX9 is mainly expressed in wheat roots, which is similar to previous research [26].

This also indicates that TaWOX9 may participate in wheat root development, and its function may be similar to that of AtWOX5 or AtWOX7. TaWUS is the putative ortholog of AtWUS and OsWOX1. In Arabidopsis, WUS is involved in the maintenance of stem cells in floral meristems, and its expression level determines the number of flower organs [27]. The high expression of TaWUS in the young spikes of wheat suggests that it may be associated with the maintenance of stem cells in young spikes and initiation in floral organs. In addition, TaWUS may participate in the development of other organs, because its expression can also be detected in other organs, as well as in young spikes. For example, TaWUS is expressed in the anthers and ovary, suggesting that it may be related to wheat fertility. Furthermore, OsWOX1 is closely related to rice fertility [16]. To confirm the predicted function of TaWOX genes, we constructed TaWUS-OE and TaWOX9-OE transgenic Arabidopsis lines. As predicted, TaWOX9 overexpression promoted Arabidopsis root development. This further confirmed that TaWOX9 plays an important role in regulating root development. The overexpression of TaWUS was able to increase the number of petals and sepals in Arabidopsis, suggesting that this gene can promote floral meristem activity. The overexpression of TaWUS also increased the number of floral buds, indicating that it can also promote inflorescence meristem activity. TaWUS is similar to AtWUS in regulating flower meristems and inflorescence meristems; however, some differences were also detected. For example, we found no increase in the number of stamens and carpels in TaWUS-OE (Figure 6B–D; Appendix A). Previous studies have shown that increased expression of AtWUS can often lead to increased numbers of inner floral whorl organs [27]. Such differences indicate that the way in which TaWUS regulates flower organ development differs from the way in which AtWUS performs a similar function. TaWUS is mainly involved in the initiation of outer floral whorl organs. Moreover, we found that TaWUS could physically interact with TaSEP2 and TaSEP3, which are orthologous to Arabidopsis SEPALLATA2 (SEP2) and SEP3 (Appendix A), respectively. In Arabidopsis, SEP2 and SEP3 are required for the development of petals [28]. Therefore, we speculate that TaWUS may interact with SEP-like genes to participate in the initiation of outer floral whorl organs.

### 3.3. Identification of Interacting Genes of TaWUS and TaWOX9 Provides Clues for Further Study of Their Functions

We identified six interacting partners of TaWUS and two partners of TaWOX9. Among the six interacting proteins of TaWUS, TaFVE encoded a WD-40 repeat-containing protein, which was orthologous to *Arabidopsis* FVE (Appendix A). The FVE protein plays important roles in determining flowering time in *Arabidopsis* [29]. Previous studies have shown that TaFVE directly interacts with multiple proteins to form multiple complexes that regulate the spike developmental process [30]. Those proteins included histone deacetylase, multiple chromatin-remodeling proteins, polycomb-group proteins, multiple flower development regulation factors, and flowering time control proteins. Therefore, TaWUS is possibly associated with multiple complexes composed of these multiple interacting proteins through interactions with TaFVE (Appendix A). These multiple complexes can participate in multiple developmental events such as vernalization, transitions in growth stages, flowering time, and the development of flower organs. We also found that the flowering time in TaWUS-OE was earlier than that in the WT. This is most likely related to the functions of these complexes. In addition, TaWUS may interact with TaFVE to participate in the development of other organs, because their transcription levels can be detected in various organs such as the leaf, stem, and leaf sheath. The specific regulatory mechanism still needs to be further elucidated. The TaFLX1 is orthologous to AtFLL1 (Appendix A). The *FLL1* gene is involved in the control of flowering time and floral organ development in *Arabidopsis* [31]. A string interaction analysis also showed that *FLL1* and *FVE* are co-expressed during the same developmental processes (Appendix A). Therefore, it is also possible that TaFLX can form complexes with TaWUS and TaFVE to participate in important developmental events, such as wheat flowering time control. TaBTB18 encodes a tramtrack and broad (BTB) domain scaffold proteins, which are orthologous to AtBT2 (Appendix A). The BTB domain-containing protein can participate in both transcriptional regulation and protein ubiquitination/degradation [32]. Whether the interaction between TaBTB18 and TaWUS forms a transcription complex or a ubiquitin complex needs further verification. A tissue expression analysis showed that *TaBTB18* was mainly expressed in the leaves, sheaths, and stems, and similarly, TaWUS showed higher expression in these tissues. The interaction node between TaBTB18 and TaWUS may be involved in the development of these tissues. TaLUX1 encodes a myb-like protein (Appendix A). *TaLUX1* is expressed in multiple tissues, especially the roots. Its interaction protein, TaWOX9, was most highly expressed in the roots. Therefore, their interaction nodes may be involved in the development of wheat roots. In this study, we identified the important proteins that can interact with TaWUS and TaWOX9, by screening the yeast library, which provided direction for a specific study of the functions of TaWUS and TaWOX9.

The ubiquitin/26S proteasome (UPS) pathway degrades ubiquitinated substrate proteins and is one of the main mechanisms employed by plants to control their growth and development. The ubiquitination process is achieved by combining the ubiquitin-activating enzyme E1, ubiquitin-conjugating enzyme E2, and ubiquitin ligase E3 [33]. TaPUB4 encodes a plant U-box E3 ligase, which is orthologous to AtPUB4 (Appendix A). In *Arabidopsis*, PUB4 is not only involved in root development, but also in the development of the SAM and anther [34,35,36]. We identified TaPUB4 through yeast two-hybrid screening, and found that it could interact with TaWUS and TaWOX9. We also found that TaPUB4 could not interact with TaWOX3 and TaWOX8 in the yeast cell (Appendix A). This indicates that TaPUB4 cannot interact with all of the TaWOX genes. Moreover, we confirmed that PUB4 in rice and *Arabidopsis* could interact with corresponding WUS orthologs in yeast (Appendix A). This suggested that these interactions may be conserved in plants. In *Arabidopsis*, PUB4 reportedly participates in regulating the maintenance of the root meristem and SAM via the same pathway as WOX9 and WUS. Tissue expression analysis showed that *TaPUB4*, *TaWOX9*, and *TaWUS* were expressed in the roots and young spikes. These findings indicate that the interaction nodes of TaPUB4-TaWOX9 and TaPUB4-TaWUS may also be involved in the development of wheat roots and young spikes, in a similar manner to PUB4, WOX5, and WUS in *Arabidopsis*. In addition, the interaction node of TaPUB4-TaWUS may be involved in the development of the leaf, sheath, stem, and anther. Overall, the interactions between TaPUB4 and TaWOX9 or TaWUS are very interesting findings from the yeast two-hybrid screening. Although further studies are still needed for verification, our research provides evidence of some of the functions of TaPUB4, TaWOX9, and TaWUS. Research on the ubiquitination of the *WOX* gene has rarely been carried out, and our research provides an indication for the direction of future studies.

## 4. Materials and Methods

### 4.1. Identification and Characterization Analysis of TaWOX Genes

First, we obtained the known WOX protein sequences of *Arabidopsis* and rice (Appendix A) from Ensembl Plants (http://plants.ensembl.org) and the National Center for Biotechnology Information (NCBI) database (https://www.ncbi.nlm.nih.gov/). Then, we used them as queries in BLASTP searches against the wheat genome database included in both Ensembl Plants and NCBI. The retrieved wheat WOX protein or cDNA sequences were used as queries to repeat the process in an iterative manner. Phylogenetic trees were constructed using the MEGA X program (https://www.megasoftware.net/download_form) and the neighbor-joining approach with 1000 bootstrap replicates.

### 4.2. Analysis of Conserved Protein Motifs

The protein sequences of each *TaWOX* gene were downloaded from the wheat database in Ensembl Plants. The HD was identified using SMART software (http://smart.embl-heidelberg.de/). The WUS-box motif was defined in a strict sense, as T-L-[DEQP]-L-F-P-[GITVL]-[GSKNTCV], and consensus TLELFPLH. The ERF-associated amphiphilic repression (EAR) motif was also defined in a strict sense, as L-[ED]-L-[RST]-L [2]. The logo diagrams used to define consensus sequences were obtained, using multiple sequence alignments for HD domains, WUS-box motifs, and EAR motifs obtained by the TEXshade software [37].

### 4.3. Planting and Collection of Plant Materials

The bread wheat variety Chinese Spring (CS) was used as the experimental plant material. The CS was planted at the Northwest Agriculture and Forestry University (108° E, 34°15′ N) in China under non-stressed, natural soil conditions. Thirty seeds per family were individually and manually planted in eight rows, 25 cm apart per 1.5 m row, with a line spacing of 15 cm. The field plots were managed according to the same methods employed for commercial production. Different vegetative tissues, such as the roots, stems, leaves, leaf sheaths, and young spikes were collected from the CS plants at the stage of pistil and stamen differentiation, and the anthers, ovaries, and glumes were collected at the heading stage. The grain tissues were collected 12, 15, and 20 days after anthesis (DAA). Specific collection methods have been previously described [38].

### 4.4. Subcellular Localization and Transcriptional Activation in the Yeast GAL4 System

The complete coding sequences (CDS) of TaWUS and TaWOX9 (Appendix A), except for the stop codon, were amplified, and subcloned into 35S::EGFP (enhanced green fluorescent protein), resulting in 35S::pTaWUS-EGFP and 35S::pTaWOX9-EGFP. A recombinant plasmid was transiently expressed in *N. benthamiana* leaf cells [36]. After 36 h, tobacco leaf cells were observed on an IX83-FV1200 confocal laser scanning microscope (Olympus, Japan). The CDS of TaWUS was divided into two parts: the N-terminal (1–333 bp), and C-terminal (334–924 bp). The CDS of TaWOX9 was also divided into two parts: N-terminal (1–309 bp), and C-terminal (310–954 bp). The full-length TaWUS, N-terminal, and C-terminal fragments were amplified from the cDNAs of CS. Similarly, the full-length TaWOX9, N-terminal, and C-terminal fragments were also amplified from the cDNAs of CS. Each amplified fragment was inserted between EcoRI and SalI in the pGBKT7 vector (Clontech) as bait, according to the manufacturer instructions of the ClonExpress^®^ II-One Step Cloning Kit (Vazyme Biotech Co., Ltd., Nanjing, China). Bait clones were transformed into the yeast strain, Y2H, with the empty prey vector, pGADT7, and co-transformed cells were diluted to a concentration gradient of 10^−0^, 10^−1^ and 10^−2^ and dropped onto SD/-Ade/-His/-Leu/-Trp medium. Transactivation activity was determined, according to cell survival on the SD/-Ade/-His/-Leu/-Trp medium. The primers used are listed in Appendix A.

### 4.5. Yeast Two-Hybrid Screen and Assay

The two-hybrid screen was performed according to the MatchmakerTM Gold Yeast Two-Hybrid System User Manual instructions, using the yeast strains, Y2H and Y187 (Clontech, Japan), as well as the two-hybrid library from different wheat tissues (Takara, Japan). Yeast single clones were selected on the solid SD/-Ade/-His/-Leu/-Trp medium and grown on the liquid SD/-His/-Leu medium. Recombinant pGADT7 plasmids with cDNA inserts were rescued, re-transformed, and confirmed by the X-α-Gal filter-lift-assay before sequencing. The sequencing results with a right open reading frame (ORF), in frame with the GAL4-AD domain, were selected to query the NCBI and Ensembl Plants databases, to identify the corresponding genes in the wheat genome. Interactions between TaWUS and TaBTB18 (TraesCS3D02G408000.1), TaFLXL1 (TraesCS2D02G231800.2), TaFVE (TraesCS3A02G250900.1), TaSEP3 (TraesCS7B02G158600.1), TaPUB4 (TraesCS3D02G527900.1), and TaSEP2 (TraesCS4D02G245200.1) were detected. The WOX domain of TaWUS (WOXD) was inserted into the pGBKT7 vector as bait, and the ORFs of other genes were inserted into the pGADT7 vector. These recombinant plasmids were co-transformed into the Y2H yeast strain, and then examined for interactions on the solid SD/-Ade/-His/-Leu/-Trp medium. Similarly, interactions between TaWOX9 and TaPUB4, and TaLUXL (TraesCS2A02G226300.2) were examined by the same method. The primers used are listed in Appendix A.

### 4.6. Bimolecular Fluorescence Complementation (BiFC) Assays

Full-length CDSs of *TaWUS*, *TaWOX9*, and candidate interacting proteins were PCR-amplified and then cloned into the plasmids 35S-SYPNE and 35S-SYPCE. The resulting clones containing TaWUS and TaWOX9 were fused to the N-terminal domain of the enhanced yellow fluorescent protein (EYFP), and candidate interacting proteins were fused to the C-terminal of the EYFP. Split-YFP fused to TaWUS, TaWOX9, and candidate interacting proteins were transiently expressed in *N. benthamiana* leaf cells as previously described [39,40]. After 36 h, tobacco leaf cells were observed on an IX83-FV1200 confocal laser scanning microscope. The primers used are listed in Appendix A.

### 4.7. Quantitative Real Time Polymerase Chain Reaction (RT-qPCR) Analysis

Total RNA was extracted from the root, stem, leaf, leaf sheath, grain, ovary, glume, anther, and young spikes with the TRIzol reagent Kit (Takara Bio, Kyoto, Japan, country), and subjected to first-strand cDNA synthesis using the PrimeScript™ RT reagent (Takara, Japan) according to the manufacturer’s protocol. Grain cDNA from 12 DAA, 15 DAA, and 20 DAA were mixed in equal amounts to form a mixed cDNA sample for further tissue expression analysis. The RT-qPCR was performed on a Q3 Real Time PCR System (Applied Biosystems, Carlsbad, CA, USA) using the SYBR Premix Ex Taq II (Takara, Kyoto, Japan), according to the manufacturer’s instructions. *TaActin* [41] was chosen as an internal control. All primers were designed using the Primer Premier 5.0 software. All individual reactions were performed in triplicate (biological replicates). The primers used are listed in Appendix A.

### 4.8. Generation of Transgenic Arabidopsis

The cDNAs of *TaWUS* and *TaWOX9*, containing the full-length CDS, were inserted between the NcoI and SpeI restriction sites of the pCAMBIA1302 vector (YouBio, Wuhan, China), respectively. Seeds of *Arabidopsis* (Col-0) were obtained from our laboratory. Both constructs were used to transform *Arabidopsis* using the floral dip method and the *Agrobacterium tumefaciens* strain GV3101 [42]. The screening, cultivation, and management of transgenic *Arabidopsis* were all conducted as previously described [43]. Homozygous lines were generated by self-fertilization. The primers used are listed in Appendix A.

### 4.9. Phenotypic Analysis of Transgenic Arabidopsis

Individual *Arabidopsis* floral structure, and floral buds and siliques were photographed using a solid microscope (Olympus, Tokyo, Japan). Morphology of the *Arabidopsis* roots and plants was photographed using a digital camera (Sony, Tokyo, Japan).

## Figures and Tables

**Figure 1 ijms-21-01581-f001:**
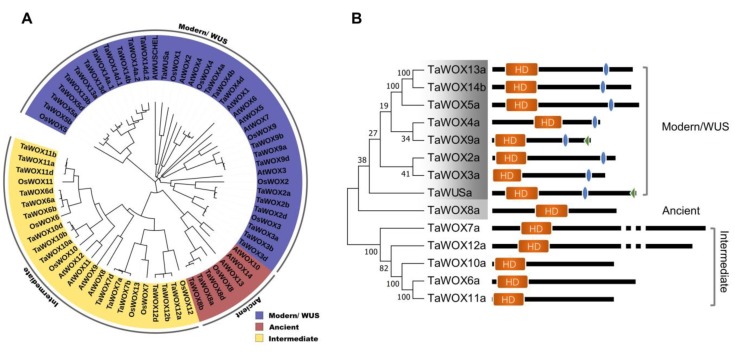
Phylogenetic relationships among WUSCHEL-related homeobox (WOX) proteins from wheat, rice, and *Arabidopsis*, and analysis of TaWOX protein domain arrangement. (**A**) Phylogenetic analysis was performed using the multiple sequence alignment generated with entire WOX protein sequences. (**B**) Depiction of the domain structure of each TaWOX protein sequence. Orange rectangles represent the homeodomain (HD). The steel blue rectangles represent the WUSCHEL (WUS)-box motif. The green triangles represent the ERF-associated amphiphilic repression (EAR)-like motif.

**Figure 2 ijms-21-01581-f002:**
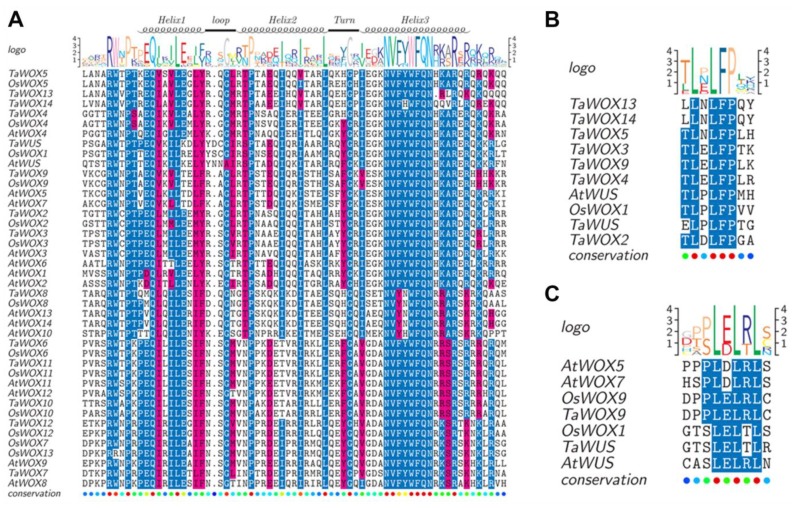
Functional domain analysis of WOX transcription factors in wheat. (**A**) Multiple sequence alignment among the homeodomain (HDs) of all WOX proteins from wheat, rice and *Arabidopsis* that carry this motif. (**B**) Multiple sequence alignment among the WUSCHEL (WUS)-box motifs of all WOX proteins from wheat, rice and *Arabidopsis* that carry this motif. (**C**) Multiple sequence alignment among the ERF-associated amphiphilic repression (EAR)-like motifs of all WOX proteins from wheat, rice and *Arabidopsis* that carry this motif. The basic sequence of the EAR-like motif was LXLXLX. Due to the high similarity of protein sequences, only the homoeologs on the A genome were selected for sequence analysis. Sequence logos represent the information content of the aligned sequences at a position in bit (max. 4.322 bit for proteins, i.e., log220) and the relative frequency of an amino acid at this position. Red font represents “similar”, blue font represents “≥50% conserved” and black front represents “non-conserved”. The color scales in the bottom consensus line represent the degree of sequence conservation (deep red represents the highest degree of conservation, and dark blue represents the lowest degree of conservation).

**Figure 3 ijms-21-01581-f003:**
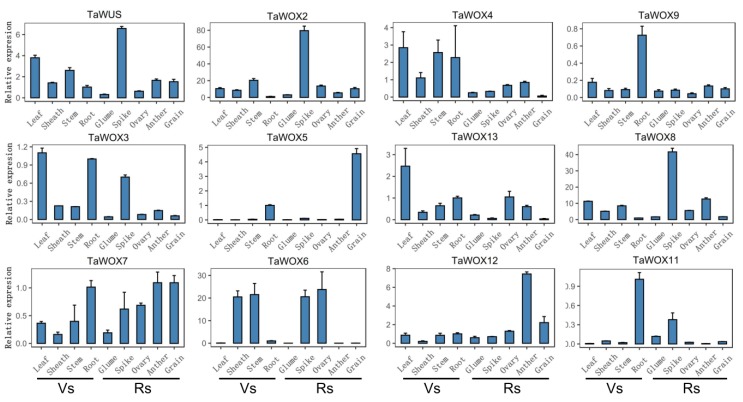
Expression profiles of 12 *TaWOX* genes in different tissues or organs. Quantitative real time polymerase chain reaction (RT-qPCR) was used to detect the expression levels of the *TaWOX* genes. The tissues or organs evaluated were the leaf, leaf sheath, stem, root, glume, spike (young spike), ovary, anther, and grain from left to right. Vs, vegetative stage. Rs, reproductive stage. *TaActin* was used as an internal control. Each value represents the mean of three biological replicates ± SE.

**Figure 4 ijms-21-01581-f004:**
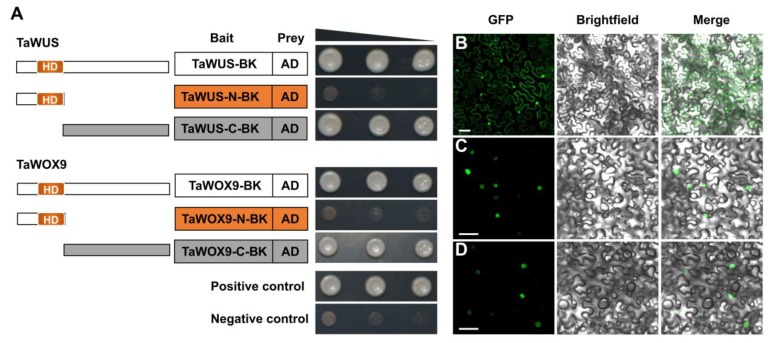
Nuclear-localized TaWUS and TaWOX9 exhibit transcriptional activity in yeast cells. (**A**) Transactivation assay of the TaWUS and TaWOX9 proteins. Full-lengths and the N- or C-terminal part of TaWUS and TaWOX9 were fused with the GAL4 DNA-binding domain and then expressed in yeast strain Y2H. The transformed yeast cells were diluted to the order of 10^−0^, 10^−1^, and 10^−2^ (gray triangle), and drops were deposited onto selective plates (SD/-Trp-His-Ade-Leu). Positive control, pGBKT7-53 and pGADT7-T vectors; Negative control, empty pGBKT7 (BK) and pGADT7 (AD) vectors. (**C**,**D**) enhanced green fluorescent protein (EGFP), TaWUS-EGFP, and TaWOX9-EGFP were transiently expressed in tobacco epidermal cells under the CaMV35S promoter. From top to bottom: EGFP (**B**), TaWUS-EGFP (**C**), and TaWOX9-EGFP (**D**). From left to right: GFP fluorescence, bright field, and merged microscope images. Scale bars = 50 μm.

**Figure 5 ijms-21-01581-f005:**
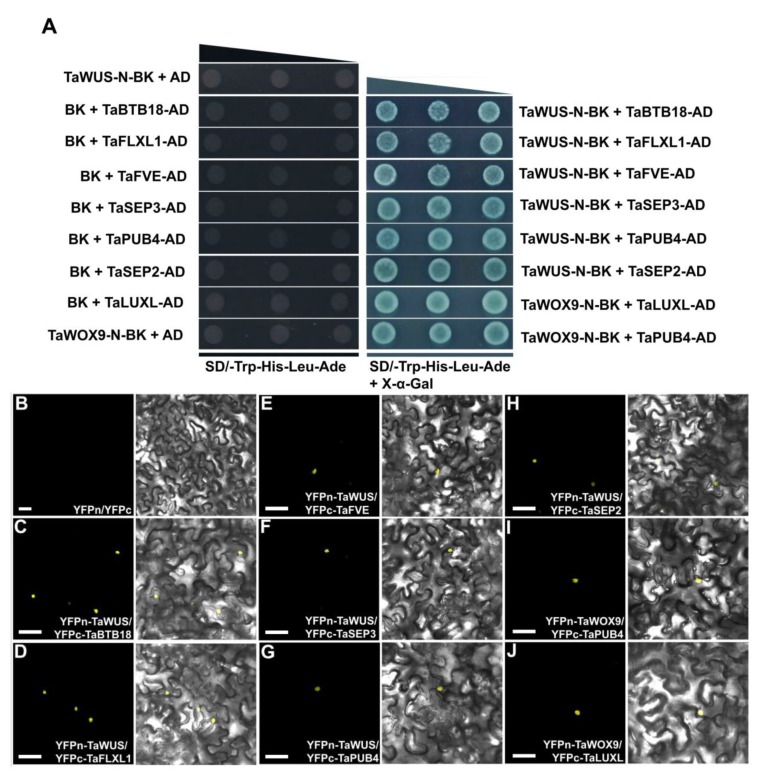
Protein interaction analysis of TaWUS and TaWOX9. (**A**) Identification of the interacting proteins of TaWUS and TaWOX9 based on yeast two-hybrid screening. Yeast co-transformed with TaWUS-N or TaWOX9-N as bait, and AD-TaBTB18, AD-TaFVE, AD-TaSEP2, AD-TaSEP3, TaPUB4, AD-TaFLXL1, or AD-TaLUXL as prey were diluted to the order of 10^−0^, 10^−1^, and 10^−2^ (gray or blue triangles), and drops were deposited onto SD/-Trp-His-Ade-Leu medium or SD/-Trp-His-Ade-Leu medium with X-α-Gal. The combination of AD-TaBTB18, AD-TaFVE, AD-TaSEP2, AD-TaSEP3, TaPUB4, AD-TaFLXL1, or AD-TaLUXL with BK, and the combination of TaWUS-N-BK or TaWOX9-N-BK with AD were used to detect self-activation. BK, pGBKT7 vector; AD, pGADT7 vector. (**B**) BiFC analyses in tobacco transient assays, where tobacco was co-transformed with YFPn and YFPc. (**C**) YFPn-TaWUS and YFPc-TaBTB18. (**D**) YFPn-TaWUS and YFPc-TaFLXL1. (**E**) YFPn-TaWUS and YFPc-TaFVE. (**F**) YFPn-TaWUS and YFPc-TaSEP3. (**G**) YFPn-TaWUS and YFPc-TaPUB4. (**H**) YFPn-TaWUS and YFPc-TaSEP2. (**I**) YFPn-TaWOX9 and YFPc-TaPUB4. (**J**) YFPn-TaWOX9 and YFPc-TaLUXL. Scale bars = 50 µm.

**Figure 6 ijms-21-01581-f006:**
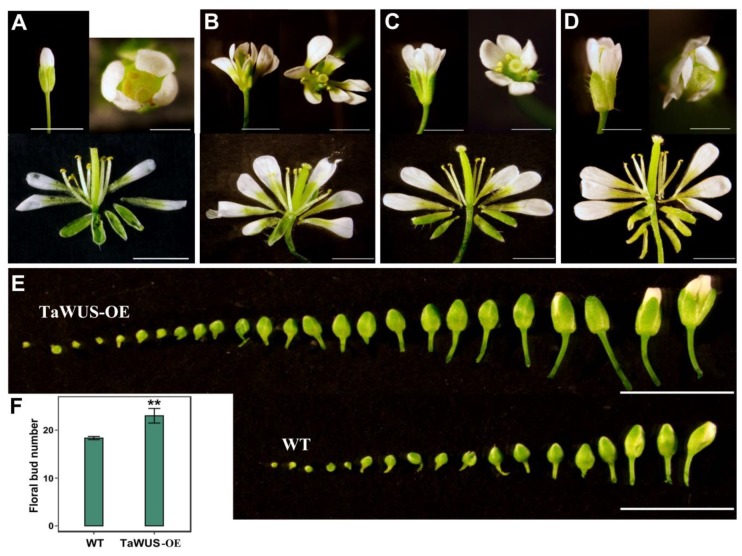
Overexpression of TaWUS (TaWUS-OE) in *Arabidopsis*. (**A**) Floral structure of wild *Arabidopsis* (WT). (**B**–**D**) Floral structures of TaWUS-OE. More petals were observed in (**B**–**D**) and more sepals in (**D**). (**E**,**F**) Inflorescence of TaWUS-OE showed a greater number of flower buds (*n* = 10), *p* < 0.001. Statistical significance was evaluated using Student’s *t*-tests. ** means very significant difference. TaWUS-OE and WT stages 7 to 12 floral buds count in 40-day-old plants (Appendix A). Scale bars = 0.5 cm (**A**–**D**); Scale bars = 1 cm (**E**). The above phenotypes were derived from two independent transgenic lines.

**Figure 7 ijms-21-01581-f007:**
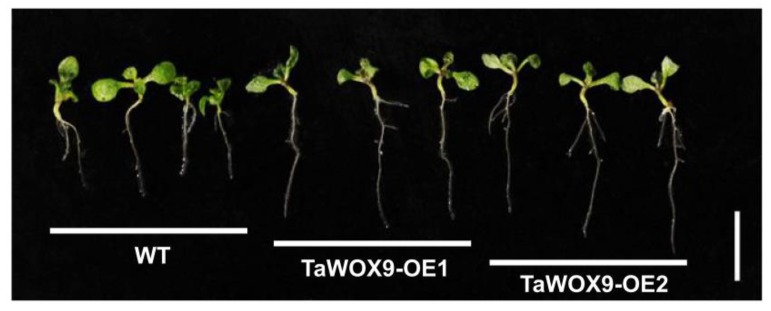
The root phenotype of overexpression of TaWOX9 (TaWOX9-OE) in *Arabidopsis* (10-day-old plants). Scale bar = 2 cm.

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
