# Peer review of "Identification of the WUSCHEL-Related Homeobox (WOX) Gene Family, and Interaction and Functional Analysis of TaWOX9 and TaWUS in Wheat"

_ijms, 2020, doi:10.3390/ijms21051581_

Round 1

Reviewer 1 Report

Li et al. use the available wheat genome sequences to study the WOX gene family in wheat. They identify the complete family members and name them in accordance with the rice WOX gene family, thereby facilitating future work on this gene family that is important for basic as well as for breeding research.

In addition to the in silico work, the authors clearly show experimentally that two of the family members, TaWUS and TaWOX9, act as transcription factors. Using Arabidopsis as heterologous system, they deliver strong hints that TaWUS and TaWOX9 have the same function in wheat as their orthologs in rice and/or Arabidopsis. They deliver also informed hypotheses on the function of other TaWOX gene family members that will facilitate future work on this important gene family.

The study is well designed and the experiments carefully conducted. The paper is generally well written, but there are paragraphs that are hard to follow and some technical terms need to be corrected. My other criticism concerns mainly the documentation of the work, in particular some missing information (see below, Major comments).

Major comments

1) There are missing information and inconsistencies on the genes and homoeologs under study:

a) There is a confusion between the homoeologs TaWOX14b and TaWOX14d. According to Figure 1a and Table S1, TaWOX14d underwent a tandem duplication, while there is only one copy of However, in the Abstract (line 19) and the Results part (lines 104 and 105), it is stated that there are two copies of TaWOX14b. Please recheck your data and adapt the manuscript accordingly.

b) In Figure 1B, only the homoeologs on the A genome are shown, and for TaWOX14 only the B genome copy. Is the domain structure of the corresponding homoeologs (and for TaWOX14 also of the paralogs) so similar, that it is sufficient to show only one of them? If yes, this interesting finding should be reported. If no, they need to be shown in a Supplementary Figure (if differences are small) or in the main text (if differences between homoeologs are big).
Similarly, in Figure 2, it is not mentioned which genome copy is depicted. Do you show the consensus sequence of the three homoeologs of each gene, or are the sequence segments identical among the homoeologs? Please report on that.
For Figure 3, see below (Point 2b).
In Figures 4, 5 and 6, results from TaWUS and TaWOX9 are presented. Which of three homoeologs of TaWUS and TaWOX was analyses in each experiment? And why did you choose them? Please report on that in the Results section.

c) In lines 96 to 99, you describe the nomenclature of the TaWOX genes. Please add an explanation on the rice OsWUS/OsWOX1 gene and state, which of the names you will use throughout you manuscript. Then go through your paper and use only one name. For example on lines 150 and 317 you use the name “OsWUS”, while “OsWOX1” is used anywhere else in your text.

d) Add a Table to the Supplementary Materials where you list all genes from rice and Arabidopsis, together with their gene identifier number from NCBI or Ensembl, that were used for the search for TaWOX genes. Refer to this Table in the Chapter 4.1 of the Materials and Methods Section.

e) In Table S1, the homoeologs TaWUSb and TaWUSd are obviously missing in the annotation of Ensembl Plants. I assume that you annotated these genes by yourselves. Therefore, publish your gene annotation in the Supplementary Materials section and submit it to the Wheat@URGI portal (https://wheat-urgi.versailles.inra.fr/Seq-Repository/Manually-curated-genes). This is important in order to make your results accessible to the community.

f) Please publish the full length cDNA sequences of TaWUS and TaWOX9 that you used for the subcellular localisation and for BiFC Assay, either in the Supplementary Materials or in a public database, for example NCBI.

g) At several places (see minor comments), there are unusual uses of technical terms (confusion with homolog and homoeologs, allohexaploidy). I strongly recommend using these terms as defined in the following paper:
Glover NM, Redestig H, Dessimoz C. Homoeologs: What Are They and How Do We Infer Them?. Trends Plant Sci. 2016;21(7):609–621. doi:10.1016/j.tplants.2016.02.005
Please recheck your manuscript if the terms used really mean what you intended to (also the ones that I suggested; see Minor Comments).

2) Chapter 2.3 which covers the topic of TaWOX gene expression, needs re-writing since it has the following shortcomings:

a) TaActin was used for normalization of all RT-qPCR reactions. However, it is well documented that the use of a single control gene may bias the results because they are insufficiently stably expressed; reviewed for example in:
Tenea GN, Peres Bota A, Cordeiro Raposo F, Maquet A. Reference genes for gene expression studies in wheat flag leaves grown under different farming conditions. BMC Res Notes. 2011;4:373. Published 2011 Sep 27. doi:10.1186/1756-0500-4-373
or in
Long XY, Wang JR, Ouellet T, Rocheleau H, Wei YM, Pu ZE, Jiang QT, Lan XJ, Zheng YL. Genome-wide identification and evaluation of novel internal control genes for Q-PCR based transcript normalization in wheat. Plant Mol Biol. 2010 Oct; 74(3):307-11. doi: 10.1007/s11103-010-9666-8
Please report which Actin among the Actin genes of wheat was used and how it was validated as control gene.

b) Each WOX gene exists in three homoeologous copies. Please report which copies you amplified by the primers you used for RT-qPCR.

c) You do not present own data on the expression of TaWOX10 and TaWOX14 in Figure 3. Please mention shortly why and move the data you retrieved from expression databases shown in Figure S3 to Figure 3.

d) The expression results are presented intricately. Please bring the results of the different genes in a logical order (Figure 3). Furthermore, do not simply describe the results of each gene, one after the other. Instead, summarize the results by tissue. Start to search for genes that are predominantly active in vegetative or reproductive organs/tissues. Then pick the most intriguing patterns (for example, TaWox9 and TaWOX11 are predominantly active in the root, while TaWOX3, TaWOX4 and TaWOX7 show strong expression in the root, but also in other tissues, etc.).

3) The phenotype of the TaWOX9 overexpression lines is presented in Figure S4B. Since this results is as important as the phenotype of TaWUS-OE (Figure 6), move the photograph in Figure S4B to Results section by creating Figure 7.

4) In the Discussion section, you present new data. It is confusing when new results (Supplementary Figures S1, S2, S3, S4 and S5) are presented in the Discussion section. Therefore, the major part of Chapters 3.3 and 3.4 need to be integrated into the Results section. Thereby, make sure you refer to Figure S3.

Minor comments

Abstract

line 18: Please replace “had homologous copies distributed on subgenomes A, B and D” by the more precise wording “had positional homoeologs on subgenomes A, B and D”.

line 26: specify the libraries: “cDNA expression libraries”

line 28: In order to enhance readability, replace “TaWOX family” by “WOX gene family in wheat”.

Introduction

line 39: The wording “based on different characteristics” has no information content. Please specify or delete.

line 45: Delete “As”.

line 72: Add the results of the study on the WOX3 gene in barley, since barley is a closer relative to wheat than rice and may give the best hint on the function of TaWOX3.
Yoshikawa T, Tanaka SY, Masumoto Y, et al. Barley NARROW LEAFED DWARF1 encoding a WUSCHEL-RELATED HOMEOBOX 3 (WOX3) regulates the marginal development of lateral organs. Breed Sci. 2016;66(3):416–424. doi:10.1270/jsbbs.16019

lines 78 and 79: Specify ‘wheat’ by writing ‘bread wheat’.

line 85: Replace “useful for future research” by a more precise and concrete statement.

Results

Lines 88-94: This text is essentially a copy of the first paragraph in the Results section of a recently published paper by some of the authors of this paper ( J. Mol. Sci. 2019, 20(17), 4319; https://doi.org/10.3390/ijms20174319 ), except that the gene names were exchanged. Please rephrase the text to avoid paraphrasing plagiarism.

line 89: The reference to the wheat genome database is missing:
The International Wheat Genome Sequencing Consortium (IWGSC). Shifting the limits in wheat research and breeding using a fully annotated reference genome. Science, Vol 361, Issue 6403, eaar7191 DOI: 10.1126/science.aar7191

line 91: exchange “heterologous hexaploid plant” by the more common description “allohexaploid plant”

line 92: replace “redundant homologous copies” by “homoeologous genes”.

Table S1: replace the column header “Location” by “Ensembl Gene Location”

line 97: replace “characteristics” by “homology”

line 100: replace “except” by “being absent on”

line 119: delete “three subclasses”

line 128: replace “protein structures and domains” by “protein domain arrangement”

line 134 – 140: Please rewrite these lines. Describe first what you want to do, how you do it and only then give the results.

line 142: Insert a line break after “ … group.”

line 157: replace “were conserved, similar to the conserved” by “were highly similar to or identical with”

Figure 2: Please explain in the legend the logo-scale (1 to 4), the blue and red shading of the amino acid (aa) sequences and the color-codes of the aa conservation (lowest lines in 2A, 2B and 2C).

lines 162-163: replace “some WOX proteins..” by “all WOX proteins from wheat, rice and Arabidopsis that carry this motif”. lines 164: replace “”some WOX proteins..” by “all WOX proteins from wheat, rice and Arabidopsis that carry this motif”.

line 199: start this paragraph (or alternatively, close the last paragraph), by establishing why you will focus for the rest of the manuscript on TaWUS and TaWOX9.

lines 206 -207: This sentence is hard to understand. Please explain more.

Figure 4: In the text (lines 199 to 209), you mention first Figures 4 B-D, and then only Figure 4A. Therefore, change the left part of the Figure (4A) with the right part (4B-D) and adapt the figure naming.
In Figure 4 A, three yeast colonies are shown per construct. Does the bar above the photographs indicate that the concentration of yeast cells decreases? Please clarify this in the legend.

line 212: replace “different portions” by “the N- or C-terminal part”

line 225: Please explain shortly why you used the N-terminal part of the WOX proteins as baits.

line 231: Add the sentence “One of the two proteins, TaPUB4, interacted with TaWUS too.”

lines 233 – 237: Please reformulate to explain clearer and avoid repetition of wording.

line 261: Please add the information which phenotypes are shown in Figure S4C-E.

Figure 6:
What is the difference among photographs B, C and D? and among G, G, and H?
Recheck scale bars, especially in Figures 6A and 6J. The bar in 6J is rather 1cm than 1 mm.
In your photograph naming, the letter G is missing. Please correct that.

line 263: Add an introductory sentence, for example “As TaWOX9 is expressed mainly in roots, we analysed the roots of the TaWOX9-OE lines.”

Materials and Methods

lines 410-420: This text is essentially a copy of the first paragraph in the Materials and Methods section of a recently published paper by some of the authors of this paper ( J. Mol. Sci. 2019, 20(17), 4319; https://doi.org/10.3390/ijms20174319 ), except that the gene names were exchanged. Please rephrase the text to avoid paraphrasing plagiarism.

line 430: replace “Chinese spring (CS) wheat” by “The bread wheat variety Chinese Spring (CS)”

lines 459, 484 and 487: replace “Janpan” by “Japan”

line 465 and lines 469-470: Add the gene identifiers of the NCBI and/or EnsemblPlants database to the corresponding genes of the 7 interacting proteins.

line 493: give a reference for pCAMBIA1302 vector (publication or company).

lines 499-506: delete.

Discussion

line 280: replace “chromosomes, other than” by “all chromosomes except”

line 282: replace “heterohexaploid” by “allohexaploid” and “a homologous copy” by “homoeologs”

lines 284 to 286: The statements in these two sentences (However, other TaWOX … gene family) are quite trivial. Please delete them. line 291-293: add a reference.

line 292: replace “WUS and WOX5 contain a conserved” by “WUS and WOX5 contain, apart from the WUS-box motif, a conserved”

line 297: delete “tow domains,”

line 300-301: Change the title to “Putative functions of TaWOX genes based on sequence homology, gene expression pattern and heterologous gene expression”

line 302: after “The TaWOX gene”, insert “family”

line 305: replace “TaWOX3 is homologous to OsWOX3” by the more precise “TaWOX3 is the putative ortholog of OsWOX3”

line 309: replace “TaWOX11 is homologous to OsWOX11” by the more precise “TaWOX11 is the putative ortholog of OsWOX11”

line 312: delete the meaningless sentence “Therefore, specific …”.

line 317: replace “TaWUS is homologous to AtWUS and…” by the more precise “TaWUS is the putative ortholog of TaWUS and…”

Reviewer 2 Report

Some suggestions are indicated in the attached file

Reviewer 3 Report

This manuscript by Li and colleagues investigates the WUSCHEL-Related Homeobox (WOX) gene family in wheat. The WOX family has been intensively studied in plants and is important in many developmental pathways. The authors first identify the orthologs of WOX genes in wheat and study the relevant protein motifs (homeodomain, WUS-box and EAR-like) in more detail. Next, an expression analysis of 12 genes in different tissues and organs is presented. Subsequently, the authors present nuclear localization, transactivation potential and protein interactors of TaWUS and TaWOX9. Finally, the two latter genes are overexpressed in a heterologous system (Arabidopsis) and the resulting phenotypes are discussed.

The work of Li et al is interesting and original as it combines a good overview of this gene family in wheat using bioinformatic techniques with functional studies of two selected genes. Their study also identifies several interactors that might be interesting candidates in follow-up studies.

Comments:

Only the BLAST tool was used to identify wheat orthologs of WOX genes using Arabidopsis protein sequences. The orthology links can be confirmed with e.g. the integrative orthology method on the PLAZA platform. This method integrates different evidences for orthology (including BLAST) and would strengthen the analysis. Figure 5B: YFPn/YFPc should be a positive control yet no signal is observed. Only one housekeeping gene (TaActin) for is used to normalize the RT-qPCR results. Best practice is to use multiple stable reference genes. The interactors of TaWUS and TaWOX9 are adequately presented and discussed in the text; but the reader would benefit from a summary table including the protein interactors, their ortholog in e.g. Arabidopsis and/or rice and the proposed role in development. This could be presented along with Supplementary Figure S2. The authors have overexpressed TaWUS and TaWox9 in Arabidopsis. I have several remarks on these results. I understand if the authors cannot address all these concerns as they want to demonstrate the potential function of these genes in a heterologous system. First, the TaWUS and TaWOX9 transgenes are introduced in wild-type plants and are not used to attempt to rescue Arabidopsis wus and wox5plants, respectively. As a result, it is difficult to attribute all phenotypic observations to the transgene alone, especially when the authors do not confirm heterologous overexpression in Arabidopsis with RT-qPCR and do not clearly indicate in how many independent lines they observed the presented phenotypes. In Supplemental Figure 4, additional phenotypes of TaWUS-OE and TaXOX9-OE are presented. In this figure, a phenotype in the pods of TaWUS-OE plants is shown that is not discussed in the text. Also, the analysis would be strengthened by the quantification of certain traits such as root length, flowering time or pod size in independent lines. Finally, the root phentyope of TaWOX9-OE plants can be studied in more detail to illustrate potential similar microscopic phenotypes as AtWOX5-OE plants: loss of differentiated columella cells and reduced gravitropism. New results regarding TaPUB4 interaction with TaWUS and TaWOX9 are presented in the discussion (3.4), these should have been presented before in the results section. Several authors are not mentioned in the author acknowledgements. A different study has investigated the role of WOX5 in wheat and relatives (Zhao et al., Gene, 2014) using bioinformatics and gene expression in different tissues. It makes sense to include this paper in the discussion.

Round 2

Reviewer 1 Report

In the revised manuscript, Li and colleagues address all concerns and comments by the three reviewers, with very few exceptions, all of minor importance (see below). The manuscript is reader-friendly and the work is very well presented and documented now.

There was a problem with accessing Table S4. It was impossible to open the file. The table should be resubmitted.

My criticism concerns the following answers to Reviewer 1:

Response 1:
In the revised version, there is still contradicting information on the paralogs of TaWOX14. In some places, it is stated that the WOX14 gene on chromosome 3B was duplicated (lines 20, 121, 122 and 123), while on other places, it is stated that WOX14 on chromosome 3D underwent duplication (line 134, line 364, Figure 1A, Figure S1 and Table S1). I assume that the Figures and Tables are correct and that you need to change the text at following places:
Line 20: replace TaWOX14b by TaWOX14d
Line 121: replace “3A and 3B” by “3A and 3D”
Line 122: replace TaWOX14b by TaWOX14d
Line 123: replace “3A and 3B” by “3A and 3D”

Response 5:
I agree that it is better to delete TaWUSb and TaWUSd. Please delete them also in Figure 1A. In Figure 1B, please replace TaWUSb by TaWUSa.

Response 19:
Lines 75-77: Please also add the reference (Yoshikawa et al 2016).

Further minor points:

Line 115: replace “orthologous” by “orthology”

Legend to Figure 7: replace “scale bars” by “scale bar”

Author Response

Response to Reviewer 1 Comments (Round 2)

Thanks again for your comments! We are impressed by your serious, professional and responsible attitude. We have further revised our manuscript, based on your suggestions.

Point 1: There was a problem with accessing Table S4. It was impossible to open the file. The table should be resubmitted.

Response 1: Sorry, we uploaded the shortcut of Table S4 by mistake in the revised supplementary materials. Therefore, we have now re-uploaded the supplemental materials containing the properly formatted Table S4.

Point 2: Response 1:

In the revised version, there is still contradicting information on the paralogs of TaWOX14. In some places, it is stated that the WOX14 gene on chromosome 3B was duplicated (lines 20, 121, 122 and 123), while on other places, it is stated that WOX14 on chromosome 3D underwent duplication (line 134, line 364, Figure 1A, Figure S1 and Table S1). I assume that the Figures and Tables are correct and that you need to change the text at following places:

Line 20: replace TaWOX14b by TaWOX14d

Line 121: replace “3A and 3B” by “3A and 3D”

Line 122: replace TaWOX14b by TaWOX14d

Line 123: replace “3A and 3B” by “3A and 3D”

Response 2: Thank you for reminding! We ignored such a detail. Please forgive our negligence! We have corrected these errors on the corresponding lines:
Line 20: The word “TaWOX14b” has been replaced by “TaWOX14d”

Line 121: The phrase  “3A and 3B” has been replaced by “3A and 3D”.

Line 122: The word “TaWOX14b” has been replaced by “TaWOX14d”

Line 123: The phrase  “3A and 3B” has been replaced by “3A and 3D”.

Point 3:  Response 5:

I agree that it is better to delete TaWUSb and TaWUSd. Please delete them also in Figure 1A. In Figure 1B, please replace TaWUSb by TaWUSa.

Response 3: Very sorry this is our negligence. We have revised Figure. 1 based on your comments.

Point 4: Response 19:

Lines 75-77: Please also add the reference (Yoshikawa et al 2016).

Response 4:  Sorry, we ignored the question! Now we have added the corresponding reference on line 77.

Point 5: Line 115: replace “orthologous” by “orthology”

Response 5: line 115: The word “orthologous” has been replaced by “orthology” on line 115.

Point 6: Legend to Figure 7: replace “scale bars” by “scale bar”.

Response 6: line 350: The phrase “scale bars”has been replaced by “scale bar” in the legend to Figure 7.

Once again, thank you very much for your comments and suggestions.

Reviewer 3 Report

The authors present an elaborate response to the issues raised by the reviewers. They incorporated some suggestions in their revised article which strengthens some claims. For most suggestions, the authors have not adapted the text but make reasonable claims why the presented approach was followed, or claim these experiment are presently being performed for future publications. Although it would have been interesting to see the results of the proposed experiments in this article, the presented experiments can reliably support the main claims of the authors. 

Author Response

Response to Reviewer 3 Comments (Round 2)

Point 1: The authors present an elaborate response to the issues raised by the reviewers. They incorporated some suggestions in their revised article which strengthens some claims. For most suggestions, the authors have not adapted the text but make reasonable claims why the presented approach was followed, or claim these experiment are presently being performed for future publications. Although it would have been interesting to see the results of the proposed experiments in this article, the presented experiments can reliably support the main claims of the authors.

Response 1: First of all, thank you very much for your recognition of our responses. Thanks again for your comments, which provided us with the opportunity to learn and improve, and also pointed out the direction for our further research.

Here we further explain our previous responses. The PLAZA platform can indeed provide different evidences for orthology analysis. However, the BLAST tool is competent for identifying WOX family genes in wheat. This is mainly because the number of WOX family genes is not large in plants and is very conserved [1]. The results of our final identification were further confirmed by the evolutionary tree constructed. For example, in the phylogenetic tree of Figure 1A, almost every TaWOX gene can be mapped to a unique OsWOX gene. Because of this, we named the TaWOX gene after the name of the corresponding OsWOX gene. And we also noticed in the analysis process that the complete protein sequence of TaWOXn is very similar with that of the corresponding OsWOXn. Because of this, it is very easy to identify the WOX gene family in the wheat genome through the BLAST tool. Moreover, our identification methods came from references to other related articles [2-6]. For the selection of internal reference genes, we have described it in detail in the first round of response.

In addition, some traits of the transgenic plants, such as flowering time and root length, are accurate, but we did not analyze them statistically. So it is reasonable to avoid statistical words when describing these traits. The similar example can refer to the previous article [7]. As for the deep phenotypic mining of TaWOX9-OE and the mutant complementation experiments on TaWOX9 and TaWUS, they are very constructive and worthy of our reference. We have considered them comprehensively and think they are very suitable to be completed in future experiments. Of course, the problems you mentioned, such as identifying transgenic plants by RT-qPCR, are indeed very necessary, and we have tried our best to supplement them.

In summary, our manuscript introduced the TaWOX gene family in detail and systematically for the first time, and initially revealed the function of related genes by identifying interaction proteins and transgenic Arabidopsis. This laid the foundation for more specific and in-depth functional research in the future. Hope you can understand us! Once again, thank you very much for your comments and suggestions.

References:

  1. Zhang, X.; Zong, J.; Liu, J.; Yin, J.; Zhang, D., Genome-Wide Analysis of WOX Gene Family in Rice, Sorghum, Maize, Arabidopsis and Poplar. Journal of Integrative Plant Biology 2010, 52, (11), 1016-1026.
  2. Xiaojian, Z. , Xiaozhu, W. , Tongjian, L. , Mingliang, J. , Xinshen, L. , & Yulan, Z. , et al. (2018). Identification, characterization, and expression analysis of auxin response factor (arf) gene family in brachypodium distachyon. Functional & Integrative Genomics.
  3. Li, M. , Wang, R. , Liu, Z. , Wu, X. , & Wang, J. . (2019). Genome-wide identification and analysis of the wuschel-related homeobox (wox) gene family in allotetraploid brassica napus reveals changes in wox genes during polyploidization. BMC Genomics, 20(1).
  4. Hao, Q. , Zhang, L. , Yang, Y. , Shan, Z. , & Zhou, X. A. . (2019). Genome-wide analysis of the wox gene family and function exploration of gmwox18 in soybean. Plants (Basel, Switzerland), 8(7).
  5. Yang, Z. , Gong, Q. , Qin, W. , Yang, Z. , & Li, F. . (2017). Genome-wide analysis of wox genes in upland cotton and their expression pattern under different stresses. BMC Plant Biology, 17(1).
  6. Singh, V. K. , Rajkumar, M. S. , Garg, R. , & Jain, M. . (2017). Genome-wide identification and co-expression network analysis provide insights into the roles of auxin response factor gene family in chickpea. Scientific Reports, 7(1), 10895.
  7. Qiao, L.; Zhang, W.; Li, X.; Zhang, L.; Zhang, X.; Li, X.; Guo, H.; Ren, Y.; Zheng, J.; Chang, Z., Characterization and Expression Patterns of Auxin Response Factors in Wheat. Frontiers in plant science 2018, 9, (1395).